# Effects of Tetrabasic Zinc Chloride as Alternative to High Doses of Zinc Oxide on Growth Performance, Nutrient Digestibility, Intestinal Morphology, Immune Function, and Gut Microbiota in Weaned Piglets

**DOI:** 10.3390/ani15213071

**Published:** 2025-10-23

**Authors:** Shuyu Peng, Jingzi Fang, Nan Zhang, Yi Chen, Yongxi Ma, Chunlin Wang

**Affiliations:** 1State Key Laboratory of Animal Nutrition, College of Animal Science and Technology, China Agricultural University, Beijing 100193, China; psy991118@126.com (S.P.); meg_1121@xj-bio.com (J.F.); zhangnan1426@163.com (N.Z.); 13725703029@163.com (Y.C.); 2Changsha Xinjia Bio-Engineeriong Co., Ltd., Changsha 410329, China

**Keywords:** tetrabasic zinc chloride, intestinal morphology, antioxidant capacity, gut microbiota, weaned piglets

## Abstract

**Simple Summary:**

Zinc, an essential trace element for piglets, has a crucial role in promoting their growth and development, as well as maintaining vital physiological functions. Currently, the prevalent zinc source additives, such as zinc oxide and zinc sulfate, typically exhibit limitations due to their low bioavailability. As a novel zinc source additive, tetrabasic zinc chloride (TBZC) possesses advantages such as relatively high bioavailability, pronounced growth-promoting effects, and superior stability, thus emerging as a potential high-quality alternative zinc source. In this study, TBZC was employed to assess the growth performance, serum indicators, and intestinal health of piglets. The results indicate that a dietary supplementation of 680 mg/kg of TBZC enhances the growth and development of 28-day-old weaned piglets, while also boosting their antioxidant capacity. Furthermore, the TBZC supplementation facilitated the colonization of beneficial bacteria while inhibiting the proliferation of harmful bacteria. The findings from this study reveal that incorporating TBZC into the diet enhances the overall health and intestinal well-being of weaned piglets. While TBZC is widely employed as a trace mineral supplement, its chronic effects on swine health require validation. This study confirms that TBZC exhibits enhanced biosafety over conventional ZnO, with longitudinal trials demonstrating negligible impacts on further growth metrics. TBZC sustains intestinal homeostasis via redox modulation and microbiota regulation, positioning it as a sustainable ZnO alternative for swine production.

**Abstract:**

This study aimed to investigate the effects of dietary supplementation with TBZC on the growth performance, diarrhea incidence, antioxidant ability, immune function, and intestinal health of weaned piglets. A total of 120 weaned piglets were randomly allocated to one of three dietary treatments with six replicate pens and eight piglets per pen: CON—a basal diet; ZnO—a basal diet with 1500 mg Zn/kg from ZnO; and TBZC—a basal diet with 680 mg Zn/kg from TBZC. Following a 42-day period of consuming the zinc-enriched diet, the piglets were switched to a basal diet for the remaining 28 days of the trial. The dietary TBZC increased the average daily feed intake of weaned piglets (ADFI) from days 1 to 14 and the average daily growth (ADG) from days 43 to 70 compared with the ZnO group (*p* < 0.05). The supplementation with TBZC decreased the acid-binding capacity compared with the ZnO group (*p* < 0.05). Moreover, dietary TBZC decreased the MDA concentration and increased the GSH-Px concentration on day 14 and increased the SOD activity on day 28 and the GSH-Px concentration on day 70 compared with the ZnO group (*p* < 0.05). Compared with the ZnO group, the dietary TBAC supplementation increased (*p* < 0.05) the relative abundance of cecal *Lactobacillus* spp. and *Blautia* spp., while decreasing *Blautia* spp. in the colonic contents; increased (*p* < 0.05) the relative abundance of *Prevotella* spp. and *Clostridium_sensu_stricto_1*; and reduced (*p* < 0.05) *Streptococcus* spp. Therefore, replacing 1500 mg/kg of ZnO with 680 mg/kg of TBZC improves growth performance and antioxidant capacity and regulates gut microbes in weaned piglets.

## 1. Introduction

The weaning period of piglets is a key production stage within the swine industry. However, weaning stress often triggers intestinal dysfunction, immune suppression, and microbial dysbiosis, leading to increased diarrhea rates, reduced growth performance, and to piglet mortality, all of which negatively affect the efficiency of pig production [1,2]. Historically, high-dose zinc oxide (ZnO) has been widely used to mitigate weaning stress by lowering the intestinal pH, inhibiting pathogen proliferation, and enhancing intestinal mucosal barrier function [3]. Nevertheless, the long-term application of high-dose ZnO poses multiple drawbacks: (1) unabsorbed zinc is excreted in feces, contributing to soil heavy metal contamination and ecological damage [4]; (2) it induces bacterial resistance genes (e.g., methicillin-resistant Staphylococcus aureus, MRSA), threatening public health security [5]; and (3) excessive zinc may inhibit the absorption of other trace elements, impairing post-weaning growth [6]. In recent years, many regions, including China, have imposed strict restrictions on the use of high-dose ZnO, driving animal feed factories to seek safer and more efficient alternatives.

The main component of tetrabasic zinc chloride (TBZC) is Zn_5_C_l2_ (OH)_8_•H_2_O, which is insoluble in water and soluble in weak acid. TBZC, as a novel zinc-based additive, demonstrates higher bioavailability and targeted functional potential due to its unique chemical structure (e.g., layered crystals and slow-release properties) [7]. Our previous study has shown that dietary TBZC improved the intestinal morphology, reduced the diarrhea rate, and changed the intestinal microflora in piglets, which positively affects the efficiency of pig production [8]. Moreover, at lower zinc doses, TBZC demonstrates stronger anti-diarrheal efficacy compared to pharmacological doses of ZnO used in weaned piglet diets. This enhanced effectiveness stems from TBZC’s superior ability to reduce intestinal permeability and prevent the disruption of barrier integrity [9]. However, there are no publicly available data on the effects of TBZC’s replacement of zinc oxide on the later growth performance of piglets. Therefore, this experiment was conducted to compare the effects of replacing 1500 mg/kg of ZnO with 680 mg/kg of TBZC on the growth performance, diarrhea rate, antioxidant ability, immune function, and intestinal health of weaned piglets and to provide a more comprehensive scientific basis for the rational use of TBZC in weaned piglets.

## 2. Materials and Methods

The experimental design and procedures used in this study were approved by the Animal Care and Use Committee of the Institute of China Agricultural University (Beijing, China) according to the Chinese Guidelines for Animal Welfare (AW30305202-1-2). This experiment was conducted in the Fengning Pig Experimental Base (Chengde, China). The ZnO and TBZC used in this study were provided by Changsha Xinjia Bio-Engineeriong Co., Ltd. (Changsha, China).

### 2.1. Animals and Experimental Designs

A total of 120 Duroc × Landrace × Yorkshire crossbred weaned piglets with an average weight of 7.12 kg (28 day age) were randomly assigned to 3 treatments based on body weight (BW) and sex, with 5 replicate pens consisting of 8 piglets (4 castrated males and 4 females per replicate pen) per treatment. The dietary treatments were corn–soybean basal diet (CON) either supplemented with 1500 mg/kg of Zn from ZnO or supplemented with 680 mg/kg of Zn from TBZC for 42 days. From day 43 to day 70, all animals were provided with same corn–soybean diet; pigs from the same treatment were housed in a single large pen due to the limitation of trial conditions. The corn–soybean diets meet requirements of 7 to 11 and 11 to 25 kg piglets recommended by the National Research Council [10] (Table 1). All piglets were housed in a temperature-controlled nursery (temperature 26~28 °C; humidity 60~70%) and had ad libitum access to feed and water. Piglets were fed twice daily at 08:30 and 15:30. The feeding behavior, fecal status, and mental status (activity levels, alertness, and responsiveness) of the piglets were observed daily, and the feeding situation of each pen was regularly monitored.

### 2.2. Growth Performance

All piglets were weighed individually at the start of the experiment, as well as on days 14, 28, 42, and 70 to calculate average daily gain (ADG). Moreover, daily feed intake was recorded every day, and average daily feed intake (ADFI) and feed conversion ratio (FCR, ADFI/ADG) were calculated. The occurrences of diarrhea were recorded at 09:00 every day from 1 to 42 days after weaning. The scoring system was applied to determine the rate of diarrhea as follows: 0 = hard feces; 1 = slightly soft feces; 2 = soft, partially formed feces; 3 = loose, semiliquid feces; and 4 = watery, mucous-like feces. Diarrhea rate was calculated according to the following equation [11]:$$Diarrhea\;rate\;(\%)=\frac{\mathrm{the}\;\mathrm{number}\;\mathrm{of}\;\mathrm{diarrhea}\;\mathrm{piglets}\;\times \;\mathrm{diarrhea}\;\mathrm{days}}{(\mathrm{the}\;\mathrm{total}\;\mathrm{number}\;\mathrm{of}\;\mathrm{piglets}\;\times \;\mathrm{experiment}\;\mathrm{days})\;}$$

### 2.3. Sample Collections

On days 14, 28, 42, and 70, six piglets in each group with a body weight closest to the average body weight were selected for the collection of blood. Blood samples were collected into 10 mL nonheparinized vacuum tubes from the front cavity vein and then centrifuged at 3000× *g* for 10 min at 4 °C to obtain serum. Serum samples were immediately stored at −20 °C for further analysis (Centrifuge 5424 R, Eppendorf, Hamburg, Germany).

Approximately 1.0 kg representative feed samples were separately taken from each phase of each treatment group and preserved at −20 °C. Fresh stool samples from each replicate were collected in sterile bags on days 12–14 and 40–42 and immediately stored at −20 °C. All the diets and fecal samples were ground to pass through a 1 mm sieve for further analysis.

On day 14, three piglets from each treatment group close to the median body weight were selected for slaughter. Samples of the intestinal segment from the middle region of the duodenum, jejunum, and ileum were collected after removing their contents and washed with saline. Then the intestinal samples were conserved in 4% paraformaldehyde for 24 h for morphological examination. The digesta of the jejunum, ileum, cecum, and colon were collected in sterile containers and immediately stored in liquid nitrogen for the microbiota community analysis. Gastric chyme was collected for pH value measurement. Jejunum mucosa samples were collected with 1.5 mL centrifuge tubes and immediately stored at −80 °C.

### 2.4. Chemical Analysis for Diet and Feces

The apparent total tract digestibility (ATTD) of crude protein (CP) was determined using the indigestible marker method using inherent acid insoluble ash (AIA) as the indigestible marker [12]. Ash and CP were analyzed using methods 942.05 and 990.03. The ATTD of the nutrients was calculated using the following equation [13]:$$\mathrm{ATTD}\;\mathrm{of}\;\mathrm{nutrients}\;(\%)\;=\;\{1–({\mathrm{AIA}}_{\mathrm{diet}}\;\times \;{\mathrm{Nutrient}}_{\mathrm{feces}})/({\mathrm{AIA}}_{\mathrm{feces}}\;\times \;{\mathrm{Nutrient}}_{\mathrm{diet}})\}\;\times \;100$$

A 100 g sample of air-dried feed was placed into a 500 mL beaker, and then we added 200 mL of deionized water. The mixture was then thoroughly mixed in a constant-temperature water bath heated to 37 °C for 20–30 min. After soaking, titration was gradually performed with 1 mol/L hydrochloric acid to pH = 4.0 (Shanghai Leici, Shanghai, China, model PHS-3C-3E-25-2F).

### 2.5. Intestinal Morphology Analysis

The preserved fixed intestinal specimens were dehydrated, cleared, and embedded in paraffin wax. The samples were sectioned at 5 µm thickness and installed on glass slides, and the paraffin sections were dewaxed and stained with hematoxylin and eosin. At least 4 well-oriented intact villi and the associated crypt depth of each section were obtained using the image processing and analyzing system (version 1; Leica Imaging Systems LTD, Cambridge, UK).

### 2.6. Analysis of Serum Antioxidant Parameters and Immune Indexes

The activity of total antioxidant capacity (T-AOC), glutathione peroxidase (GSH-Px), superoxide dismutase (SOD), malondialdehyde (MDA), Catalase (CAT), immunoglobulin A (IgA), immunoglobulin G (IgG), and immunoglobulin M (IgM) concentration in serum was measured using enzyme-linked immunosorbent assay (ELISA) following the manufacturer’ instructions (Nanjing Jiancheng Bioengineering Institute, Nanjing, China). All commercial ELISA kits used were specifically designed and validated for porcine samples by the manufacturer. Each assay included a standard curve, and both positive and negative controls were run alongside the samples as per the kit protocols.

### 2.7. Intestinal Microbiota Community

Approximately 0.25 g of digesta of jejunum, ileum, cecum, and colon was used to extract total bacterial DNA using a DNA kit (Omega Bio-Tek, Norcross, GA, USA). The quality and concentration of DNA were determined by 1.0% agarose gel electrophoresis and a NanoDrop^®^ ND-2000 spectrophotometer (Thermo Scientific Inc., Waltham, MA, USA) and kept at −80 °C prior to further use. The hypervariable region V3-V4 of the bacterial 16SrRNA gene was amplified using universal primers 338F (5′-ACTCCTACGGGAGGCAGCAG-3′) and 806R (5′-GGACTACHVGGGTWTCTAAT-3′). The amplification cycle conditions were as follows: 95 °C pre-denaturation for 3 min, followed by 27 cycles of 95 °C denaturation for 30 s, 55 °C annealing for 30 s and 72 °C extending for 30 s, and final extension at 72 °C for 10 min. The sequencing was performed using Illumina’s Miseq PE300 platform (Illumina, San Diego, CA, USA). The raw sequences were sequenced using fastp (https://github.com/OpenGene/fastp, version 0.19.6, accessed on 12 July 2025) software and FLASH (https://ccb.jhu.edu/software/FLASH/, version 1.2.11, accessed on 12 July 2025) software for quality control and splicing. The splicing was performed for subsequent analysis. Then, using the UPARSE software (http://drive5.com/uparse/, version 7.1, accessed on 12 July 2025), the quality control spliced sequences were subjected to operational taxonomic unit (OTU) removal of chimeras and standard clustering (with 97% confidence level) based on 97% similarity; then the representative OTU sequences were selected for annotation. Finally, the composition of microorganisms in the gut contents was analyzed based on the standardized OTUs. The grouped samples were analyzed for differences in species composition and characteristic flora using Mothur software (http://www.mothur.org/wiki/Calculators, version 1.48.0, accessed on 12 July 2025), and the results were presented in pictures.

### 2.8. Statistical Analysis

The UNIVARIATE procedure (one-way ANOVA) of SAS 9.4 (SAS Inst. Inc., Cary, NC, USA) was used to check the normality of residuals and equal variances. Differences in diarrhea rates were analyzed with the pen (replicate) as the experimental unit. Individual piglets were used as experimental unit to analyze nutrient digestibility, intestinal morphology, serum parameters, and gut microbiota. Data were analyzed using the GLM procedure of SAS followed by Tukey’s test, and the results were presented as mean values ± SEM. *p* < 0.05 indicates a significant difference; 0.05 < *p* < 0.1 indicates a trend toward significance.

## 3. Results

### 3.1. Growth Performance and Diarrhea Rate

The growth performance and diarrhea rate of weaned piglets are shown in Table 2. The ADFI from days 1 to 14 was significantly increased (*p* < 0.05) in the TBZC group compared with the CON and ZnO groups. The ADG in the TBZC group was higher (*p* < 0.05) than that in the CON and ZnO groups from days 43 to 70. However, no significant difference was observed in the diarrhea rate of weaned piglets among the three groups.

### 3.2. The ATTD of Nutrients, the Acid-Binding Capacity, and the pH Value of the Stomach

As shown in Table 3, no significant difference was observed in the ATTD of CP of weaned piglets among the different groups. The dietary ZnO supplementation group had a higher (*p* < 0.05) pH of stomach contents and acid-binding capacity compared with the CON group. In addition, the dietary TBZC supplementation decreased (*p* < 0.05) the acid-binding capacity compared with the ZnO group.

### 3.3. Intestinal Morphology

As shown in Table 4, no significant difference was observed in villus height, the crypt depth, and the ratio of the villus height to crypt depth of the duodenum, jejunum, and ileum among the three groups.

### 3.4. Serum Antioxidant Parameters

The serum MDA concentration in the TBZC group was lower than that in the CON and ZnO groups on day 14, while at days 28, 42, and 70, both the TBZC and ZnO groups showed lower levels compared to the CON group (Table 5, *p* < 0.05). The serum SOD concentration in the TBZC group was higher than that in the CON and ZnO groups on day 28 but was only higher than the CON group on days 14 and 42 (*p* < 0.05). The serum GSH-Px concentration was higher in the TBZC group compared with the CON and ZnO groups on days 14 and 70 (*p* < 0.05). The serum CAT concentration was higher in the TBZC group compared with the CON group on days 14, 42, and 70 (*p* < 0.05).

### 3.5. Serum Immune Indexes

No differences were observed among treatments in terms of serum concentrations of IgA, IgM, and IgG (Table 6).

### 3.6. Gut Microbiota Community Diversity

The analysis of OTUs in the digesta of the jejunum, ileum, cecum, and colon is shown in the Venn diagram (Figure 1). In the digesta of the jejunum, a total of 320 OTUs were shared in the three treatment groups; the CON piglets had 259 unique OTUs, the ZnO piglets had 67 unique OTUs, and the TBZC piglets had 87 unique OTUs. In the digesta of the ileum, a total of 102 OTUs were shared in the three treatment groups; the CON piglets had 133 unique OTUs, the ZnO piglets had 54 unique OTUs, and the TBZC piglets had 31 unique OTUs. In the digesta of the cecum, a total of 355 OTUs were shared in the three treatment groups; the CON piglets had 202 unique OTUs, the ZnO piglets had 363 unique OTUs, and the TBZC piglets had 198 unique OTUs. In the digesta of the colon, a total of 458 OTUs were shared in the three treatment groups; the CON piglets had 271 unique OTUs, the ZnO piglets had 229 unique OTUs, and the TBZC piglets had 327 unique OTUs. The principal coordinate analysis (PCoA) showed significant clustering characteristics of the microbial composition in the jejunum, ileum, cecum, and colon among the different groups (Figure 2).

As shown in the relative abundance bar charts (Figure 3), at the phylum level, Firmicutes and Cyanobacteria were the dominant phyla in the jejunal microbiota, followed by Actinobacteriota and Proteobacteria; Firmicutes and Actinobacteriota were the dominant phyla in the ileal microbiota, followed by Proteobacteria and Cyanobacteria; and Firmicutes and Bacteroidota were the dominant phyla in the colonic microbiota and cecum microbiota, followed by Actinobacteriota and Proteobacteria. As shown in Figure 4, at the genus level, *Streptococcus* spp. and *Lactobacillus* spp. were the dominant species in the jejunal microbiota, followed by *Clostridium_sensu_stricto_1* and *norank_f__norank_o__Chloroplast* (sequences that could not be fully classified at the order and family levels were assigned the standardized label); *Streptococcus* spp. and *Lactobacillus* spp. were the dominant species in the ileal microbiota, followed by *Clostridium_sensu_stricto_1* and *Terrisporobacter* spp.; and *Lactobacillus* spp. and *Clostridium_sensu_stricto_1* were the dominant species in the colonic microbiota and cecum microbiota, followed by *Terrisporobacter* spp. Compared with the ZnO group, the dietary TBZC supplementation increased (*p* < 0.05) the relative abundance of cecal *Lactobacillus* spp. and *Blautia* spp., while decreasing (*p* < 0.05) *Blautia* spp. in the colonic contents; increased (*p* < 0.05) the relative abundance of colonic *Prevotella* spp. and *Clostridium_sensu_stricto_1*; and reduced (*p* < 0.05) *Streptococcus* spp.

## 4. Discussion

Improving the performance and preventing or reducing the diarrhea rate of weaned piglets have always been focuses of the pig industry. In addition to ensuring a suitable feeding environment, the timely administration of vaccines, and the implementation of other production techniques, the selection of appropriate feed additives also helps to prevent and treat diseases, thereby enhancing the resistance of piglets [14]. The addition of ZnO to feed has become a common measure to control diarrhea in piglets after weaning and to promote their growth [15,16,17]. However, due to the potential toxic side effects of ZnO, a global decision has been made to phase out the excessive use of zinc oxide in piglet diets. Therefore, this study aimed to investigate the effects of adding TBZC instead of ZnO on the health of piglets. Our results showed that TBAC in the diet can improve the growth performance, antioxidant ability, and gut microbiota of weaned piglets.

It was found that adding 3000 mg/kg of ZnO to the diet of weaned piglets maintained plasma zinc levels between 1.5 and 3.0 mg/L, resulting in optimal ADG for the piglets [18]. However, a review of the current literature reveals that high-dose ZnO supplementation has multiple toxic side effects. These mainly include inhibited growth and development and impaired organ development [6]. A high-zinc diet can increase intestinal metallothionein contents, raising the likelihood of copper binding and significantly reducing copper absorption by the body [19]. Moreover, high-dose zinc causes excessive accumulation in tissues like the liver, kidney, and pancreas of piglets, resulting in zinc overload. Refs. [6,20] reported that adding 1000 mg/kg of TBZC to the diet as a zinc source instead of ZnO can achieve feeding effects comparable to high-dose ZnO regarding growth performance and diarrhea control. Our results found that TBZC significantly increased the ADFI from days 0 to 14 and the ADG from days 42 to 70 compared with the CON group and ZnO groups, which is consistent with the previous study. Furthermore, compared to the ZnO group, TBZC supplementation exhibited lower toxicity and did not show significant antagonistic effects on the posterior growth performance of piglets, thereby offering a safer alternative to zinc oxide. However, dietary TBZC did not improve the diarrhea rate of piglets compared with the CON and ZnO groups in the present study, which is inconsistent with previously reported results [20]. This may be attributable to differences in the amount of TBZC added and piglets in the two studies.

ZnO is considered as a feed additive with a high acid-binding capacity and reacts strongly in acidic environments. Since it disrupts the stomach acid environment, reduces digestive enzyme activity, and affects digestion and absorption in animals [21], it is often necessary to add additional organic acids, such as acidifiers, to counteract its potential side effects [8]. Compared with ZnO, TBZC exhibits more stable chemical properties and is only partially dissolved in the stomach acid environment, and most of TBZC enters the intestine in a molecular state and exerts its effects in the intestine, which can effectively reduce the waste of zinc. The present study showed that a lower dietary acid-binding capacity was observed in the TBZC group compared with the ZnO group, which reduced the influence of feed digestion on the gastric acid–alkali environment. Furthermore, the PH value of the gastric digesta was lower in the TBZC group compared with the ZnO group, which also confirmed the above results.

Intestinal histomorphology has been widely used for assessing intestinal development and function. The decreased digestion and absorption of nutrients due to villous atrophy and crypt hypertrophy as a result of early weaning may contribute to diarrhea [22]. The present study found no statistic differences in the villus height and crypt depth of the ileum of piglets, which is consistent with the previous study, indicating that zinc did not significantly affect the ileum morphology [23]. However, dietary TBZC increased the ratio of the villus height to crypt depth in the duodenum and jejunum to a certain extent compared with the ZnO group (8.74–56.56%), which also explained the growth-promoting effect of TBZC.

The production and elimination of free radicals in piglets represent a dynamic redox equilibrium, with the maintenance of this homeostasis being crucial for normal physiological functions. In this study, we found that the serum MDA decreased and the serum SOD, GSH-Px, and CAT increased with the inclusion of TBZC in their diets. As a byproduct of the lipid peroxidation from polyunsaturated fatty acids, MDA is an important indicator of the degree of oxidative stress in weaned piglets [24]. The three antioxidant enzymes, SOD, GSH-Px, and CAT, through their synergistic effects, effectively reduce the generation and accumulation of harmful substances such as superoxide anions, hydroperoxide, and hydrogen peroxide and thus relieve the oxidative stress faced by cells [25]. Consequently, the observed results suggest that dietary TBZC enhances the antioxidant capacity of weaned piglets compared with ZnO.

Zinc deficiency impairs the phagocytic and cytotoxic capabilities of neutrophils, macrophages, and natural killer (NK) cells, reducing the efficiency of pathogen clearance. Moreover, zinc enhances both innate and adaptive immunity through multiple pathways and maintains normal serum levels of IgG, IgM, and IgA. Refs. [3,26] reported that IgM and IgG levels in piglets were upregulated by the addition of ZnO and nano-ZnO. However, the results showed that the administration of two different zinc compounds failed to cause significant alterations in serum IgA, IgM, and IgG profiles in the present study. These results are also inconsistent with the previous findings [8], which showed that dietary supplementation with 0.8 to 1.2 g/kg of TBZC improved ileal IgA concentrations. Therefore, we speculate that this may be attributed to differences in the amount of TBZC added.

The piglet intestine hosts a microflora that interacts symbiotically with the host. This microflora constitutes the microecological balance in the piglet intestine, which is essential for various physiological processes, including growth and development, disease resistance, and nutrient metabolism. Consequently, the research on the intestinal flora of weaned piglets has attracted increasing attention in the scientific community [27]. This study systematically characterized the distinct modulatory effects of ZnO and TBZC on the composition and spatial distribution of the intestinal microbiota in weaned piglets. At the phylum level, both zinc sources induced a significant enrichment of Cyanobacteria in jejunal contents, while the cecal microbiota showed differential responses: Bacteroidota was consistently elevated, and Actinobacteriota and Proteobacteria were markedly reduced in both treatment groups. Further genus-level analysis revealed specific responses to the TBZC supplementation: cecal *Lactobacillus* spp. and *Blautia* spp. abundances were significantly upregulated, whereas colonic Blautia spp. were decreased. Concurrently, colonic *Prevotella* spp. and *Clostridium_sensu_stricto_1* were significantly increased, accompanied by a marked reduction in *Streptococcus* spp. These multi-level microbial shifts suggest that zinc sources may exert their beneficial effects through specific mechanisms. The enrichment of Bacteroidota and the reduction in Firmicutes in cecal contents may reflect the promoting effect of TBZC on carbohydrate metabolism. Bacteroidota-dominated microbiota are often associated with the efficient degradation of dietary fiber, and their metabolites, such as SCFAs, enhance intestinal barrier function [28]. The decreased abundance of Firmicutes may reduce the host energy uptake. In addition, the significant reduction in Proteobacteria may be attributed to the broad-spectrum antimicrobial activity of zinc, especially against Gram-negative bacteria such as *Escherichia coli* spp. [29]. However, the rise in this phylum in the cecum should alert us to the proliferation risk of potential pathogenic bacteria. The significant increase in *Lactobacillus* spp. in the cecum was one of the important findings in the TBZC group. As a classic probiotic, *Lactobacillus* spp. can reduce the intestinal pH and inhibit pathogen colonization by producing lactic acid [30]. The increase in its abundance may be directly related to the selective inhibition of zinc in pathogens and the microenvironment support for probiotics. Meanwhile, the enrichment of *Blautia* spp. in the cecum is noteworthy: this genus is able to use acetic acid to produce butyric acid [31], which is the main energy source for colonic epithelial cells and may alleviate weaning stress by enhancing barrier integrity. However, the decreased abundance of *Blautia* spp. in the colon contrasts with the increased abundance of *Prevotella* spp., a core fiber-degrading bacterium of Bacteroidota, which may indicate that TBZC promotes the fermentation capacity of dietary fiber in the posterior segment of the colon. The significant increase in *Clostridium_sensu_stricto_1* in the colon warrants cautious interpretation. Although certain Clostridial species (e.g., *Clostridium butyricum*) exhibit probiotic properties, *Clostridium_sensu_stricto_1* includes spore-forming pathogens (e.g., *C. perfringens*) whose overgrowth may disrupt intestinal mucosal barrier function and increase intestinal permeability [32]. Conversely, the decline in *Streptococcus* spp. may represent a positive outcome, as certain strains (e.g., S. suis) are associated with intestinal inflammation in piglets [33]. These conflicting results highlight the need for the integrated assessment of zinc supplementation effects using metabolomic profiling and host inflammatory markers to fully evaluate its comprehensive impact on gut microbiota–host interactions.

## 5. Conclusions

This study demonstrates that supplementing weaned piglets with 680 mg/kg of TBZC improves growth performance, enhances the antioxidant capacity, and regulates the gut microbiota. These multifaceted benefits yield a superior long-term growth performance compared to ZnO, thereby positioning TBZC as a viable high-efficacy alternative.

## Figures and Tables

**Figure 1 animals-15-03071-f001:**
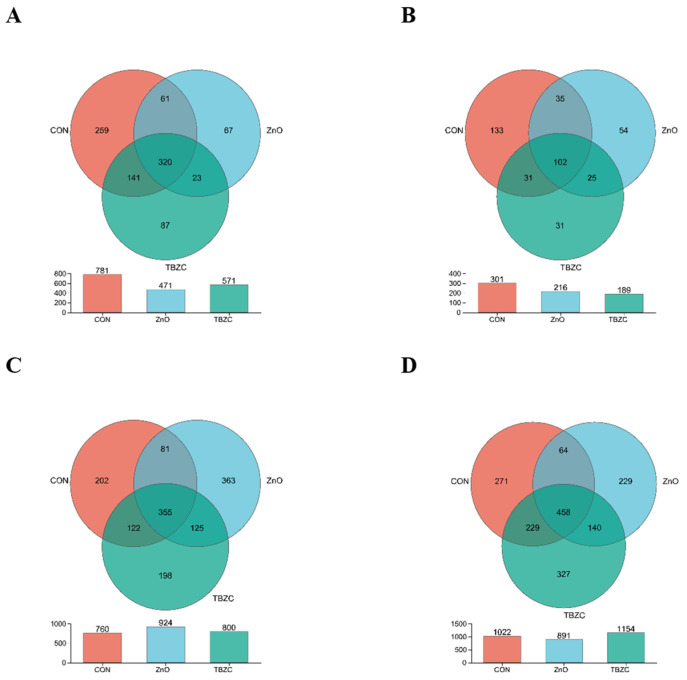
The bacterial operational taxonomic unit community composition of the jejunum, ileum, cecum, and colon (represented by (**A**–**D**)) in piglets. Venn diagrams of the bacterial operational taxonomic unit communities among three treatment groups: control diet (CON), control diet + 1500 mg/kg ZnO (ZnO), and control diet + 680 mg/kg of TBZC (TBZC).

**Figure 2 animals-15-03071-f002:**
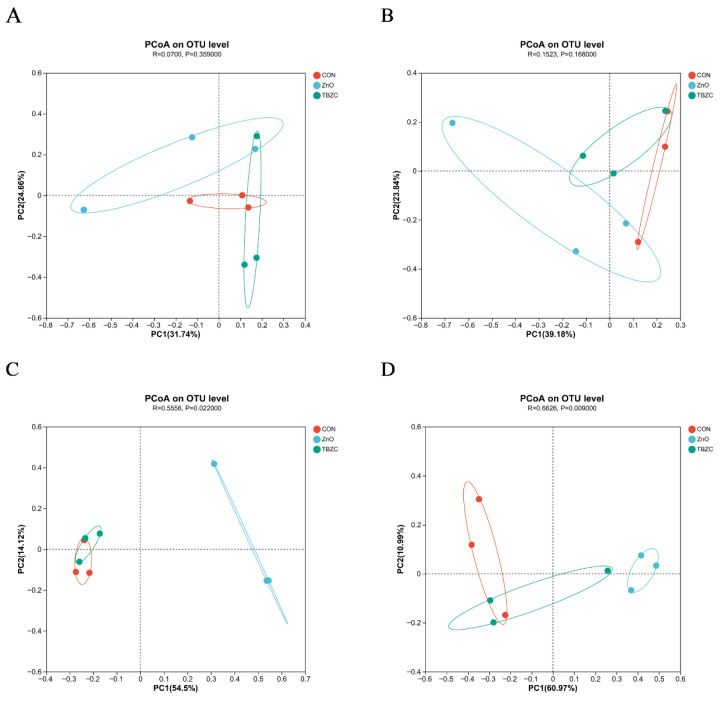
Effects of zinc treatment of analysis of intestinal microbial β diversity of piglets. Jejunum, ileum, cecum, and colon are represented by (**A**–**D**), respectively.

**Figure 3 animals-15-03071-f003:**
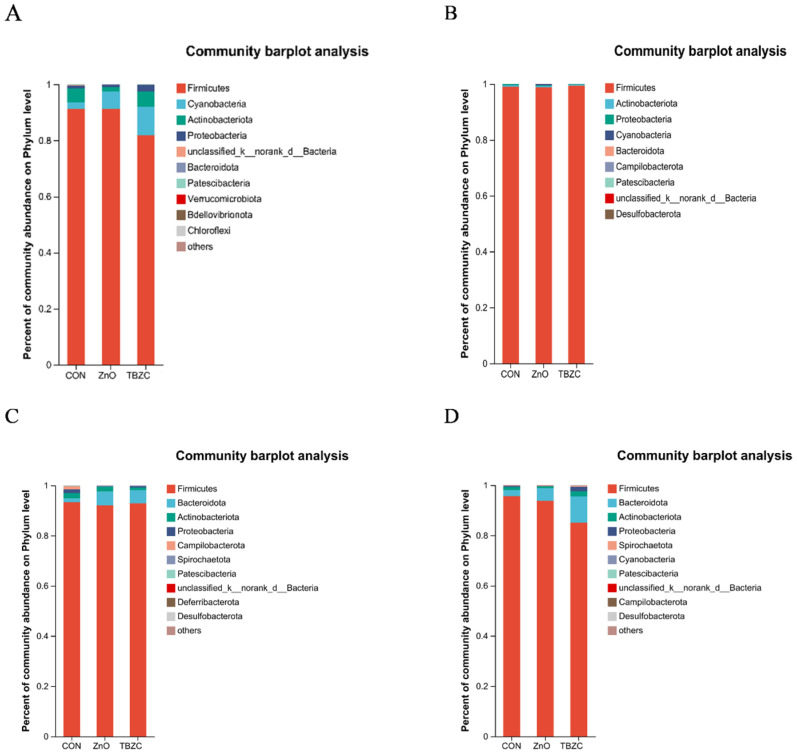
Effects of zinc treatment on of the jejunum, ileum, cecum, and colon (represented by (**A**–**D**)) bacterial communities of piglets at the phylum level.

**Figure 4 animals-15-03071-f004:**
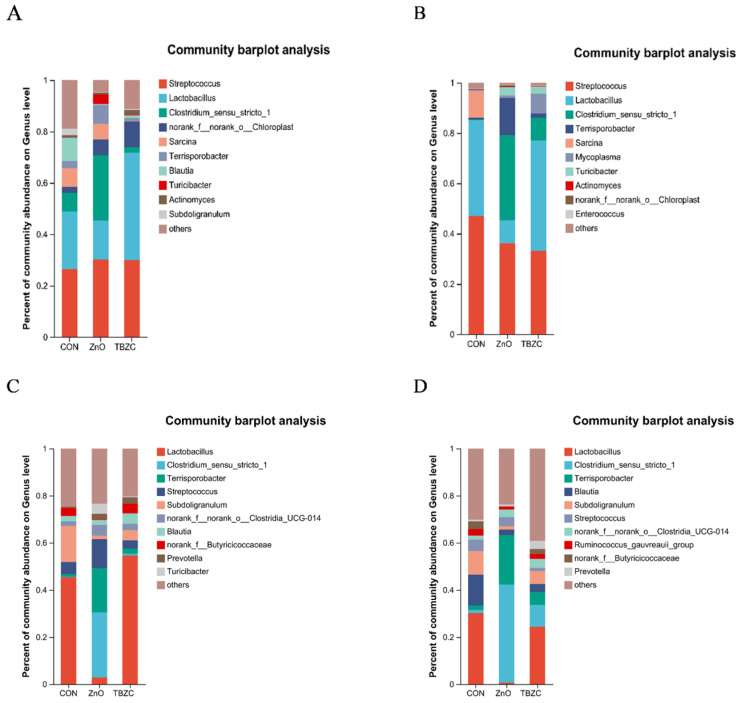
Effects of zinc treatment on of the jejunum, ileum, cecum, and colon (represented by (**A**–**D**)) bacterial communities of piglets at the species level.

**Table 1 animals-15-03071-t001:** Ingredient composition of the basal diets (%, as-fed basis).

Ingredients	Period 1 (d 1 to 14)	Period 2 (d 15 to 42)	Period 3 (d 43 to 70)
Corn	62.12	68.10	68.10
Extruded soybeans	6.00	5.00	5.00
Peeled soybean meal	13.00	12.00	12.00
Fermented soybean meal	3.50	2.50	2.50
Fish meal	4.00	3.20	3.20
Whey powder	3.00	2.00	2.00
Glucose	4.00	3.00	3.00
Dicalcium phosphate	1.20	1.20	1.20
Limestone	0.70	0.60	0.60
Salt	0.35	0.35	0.35
Soy oil	1.00	1.00	1.00
L-Lys HCI, 78%	0.40	0.35	0.35
DL-Methionine, 98%	0.09	0.08	0.08
L-Threonine, 98%	0.10	0.09	0.09
L-Tryptophan, 98%	0.04	0.03	0.03
Premix ^1^	0.50	0.50	0.50
Nutrient levels ^2^			
Digestible energy, kcal/kg	3410	3410	3410
Crude protein, %	18.70	17.20	17.20
Total calcium, %	0.71	0.63	0.63
Total phosphorus, %	0.55	0.52	0.52
Lysine, %	1.38	1.22	1.22
Methionine, %	0.41	0.37	0.37
Threonine, %	0.78	0.72	0.72
Tryptophan, %	0.24	0.22	0.22

^1^ The premix provided the following (per kilogram of compound feed): Vitamin A, 12,000 IU; Vitamin D3, 2000 IU; Vitamin E, 30 IU; Vitamin K3, 3.0 mg; Vitamin B1, 3.0 mg; Vitamin B2, 10 mg; Vitamin B6, 6.0 mg; Vitamin B12, 24 μg; nicotinic acid, 30 mg; d-pantothenic acid, 30 mg; folic acid, ^2^ mg; biotin, 0.3 mg; choline chloride, 600 mg; Fe, 120 mg; Cu, 10.0 mg; Zn, 110 mg; Mn, 35 mg; I, 0.3 mg; and Se, 0.3 mg.

**Table 2 animals-15-03071-t002:** Effects of TBZC on performance and diarrhea rate in piglets ^1^.

Items	CON ^1^	ZnO	TBZC	SEM ^2^	*p*-Value
0 d BW, kg	7.12	7.12	7.12	0.25	1.00
14 d BW, kg	11.08	10.92	11.41	0.20	0.61
28 d BW, kg	17.50	17.12	17.61	0.31	0.80
42 d BW, kg	25.61	25.74	26.18	0.42	0.85
70 d BW, kg	41.25	38.57	42.75	0.75	0.07
Days 1 to 14					
ADG, g	269.82	271.69	305.36	13.22	0.50
ADFI, g	393.39 ^b^	390.89 ^b^	421.43 ^a^	18.25	0.02
FCR, g/g	1.46	1.44	1.38	0.02	0.27
Diarrhea rate, %	2.32	1.25	0.89	0.61	0.12
Days 15 to 28					
ADG, g	456.87	431.53	442.3	14.77	0.80
ADFI, g	696.96	658.36	675.83	20.91	0.85
FCR, g/g	1.55	1.54	1.54	0.06	0.99
Diarrhea rate, %	0.60	0.20	0.19	0.20	0.44
Days 29 to 42					
ADG, g	506.38	533.78	535.46	11.29	0.53
ADFI, g	1076.52	1022.68	1039.38	34.04	0.84
FCR, g/g	2.13	1.91	1.93	0.05	0.27
Diarrhea rate, %	0.99	0.20	0.39	0.34	0.19
Days 43 to 70					
ADG, g	558.33 ^a^	458.16 ^a^	591.79 ^b^	14.83	<0.01
ADFI, g	1248.16	1143.98	1244.69	*	*
FCR, g/g	2.23	2.49	2.10	*	*

BW, body weight; ADG, average daily gain; ADFI, average daily feed intake; F/G, ratio of feed to gain. ^1.^ CON = piglets in the control group fed a basal diet; ZnO = piglets in ZnO group fed a basal diet supplemented with 1500 mg/kg of ZnO; TBZC = piglets in TBZC group fed a basal diet supplemented with 680 mg/kg of TBZC. ^2.^ SEM, standard error of the mean (*n* = 5). a, b Mean values within a row with different letters differ at *p* < 0.05. And 0.05 ≤ *p* < 0.10 was considered as a tendency. * Due to the group-housing of all treatment groups from day 43 to day 70 of the experiment, the resulting insufficient number of replicate samples within each group precluded the estimation of variability, thereby rendering the calculation of *p*-values and standard errors unfeasible.

**Table 3 animals-15-03071-t003:** Effects of TBZC on CP digestibility, acid-binding capacity, and pH of stomach contents in piglets ^1^.

Items	CON ^1^	ZnO	TBZC	SEM ^2^	*p*-Value
ATTD of CP					
d 14	77.51	77.25	74.69	0.72	0.22
d 42	72.66	72.67	73.39	0.92	0.94
pH of stomach contents	3.40 ^b^	4.56 ^a^	4.13 ^ab^	0.21	0.05
Acid-binding capacity	15.40 ^c^	19.36 ^a^	16.37 ^b^	0.44	<0.01

^1^ CON, piglets in the control group fed a basal diet; ZnO, piglets in ZnO group fed a basal diet supplemented with 1500 mg/kg of ZnO; and TBZC, piglets in TBZC group fed a basal diet supplemented with 680 mg/kg of TBZC. ^2^ SEM, standard error of the mean (*n* = 3). ^a, b, c^ Mean values within a row with different letters differ at *p* < 0.05. And 0.05 ≤ *p* < 0.10 was considered as a tendency.

**Table 4 animals-15-03071-t004:** Effects of TBZC on small intestinal morphology in piglets ^1^.

Items	CON ^1^	ZnO	TBZC	SEM ^2^	*p*-Value
Duodenum					
Villus height, μm	331.04	424.81	543.03	41.13	0.13
Crypt depth, μm	355.25	436.83	539.37	38.17	0.16
Villus height/crypt depth	0.71	1.03	1.12	0.08	0.56
Jejunum					
Villus height, μm	389.24	473.70	540.92	32.75	0.12
Crypt depth, μm	286.97	353.07	295.84	10.68	0.41
Villus height/crypt depth	1.23	1.22	1.91	0.19	0.22
Ileum					
Villus height, μm	269.63	286.94	302.15	13.99	0.76
Crypt depth, μm	303.68	364.32	359.06	17.93	0.36
Villus height/crypt depth	0.95	0.88	0.94	0.08	0.92

^1^ CON, piglets in the control group fed a basal diet; ZnO, piglets in ZnO group fed a basal diet supplemented with 1500 mg/kg of ZnO; and TBZC, piglets in TBZC group fed a basal diet supplemented with 680 mg/kg of TBZC. ^2^ SEM, standard error of the mean (*n* = 3).

**Table 5 animals-15-03071-t005:** Effects of TBZC on serum antioxidant parameters in piglets ^1^.

Items	CON ^1^	ZnO	TBZC	SEM ^2^	*p*-Value
d 14					
MDA, nmol/mL	3.82 ^a^	3.78 ^a^	3.06 ^b^	0.11	<0.01
SOD, U/mL	53.76 ^b^	58.73 ^ab^	64.29 ^a^	1.50	<0.01
GSH-Px, U/mL	111.62 ^b^	126.97 ^b^	146.88 ^a^	4.53	<0.01
CAT, U/mL	34.17 ^b^	40.02 ^ab^	46.32 ^a^	1.65	<0.01
T-AOC, U/mL	7.05	7.83	9.58	0.48	0.08
d 28					
MDA, nmol/mL	3.75 ^a^	2.82 ^b^	2.77 ^b^	0.14	<0.01
SOD, U/mL	68.39 ^b^	66.59 ^b^	77.70 ^a^	1.81	0.01
GSH-Px, U/mL	139.24	150.31	164.86	5.07	0.11
CAT, U/mL	43.50	44.51	48.88	1.11	0.10
T-AOC, U/mL	8.59	8.68	9.09	0.14	0.34
d 42					
MDA, nmol/mL	4.02 ^a^	2.66 ^b^	2.39 ^b^	0.22	<0.01
SOD, U/mL	74.29 ^b^	79.03 ^ab^	82.41 ^a^	1.31	0.02
GSH-Px, U/mL	153.18	163.06	178.46	5.92	0.22
CAT, U/mL	45.51 ^b^	48.78 ^ab^	54.24 ^a^	1.42	0.02
T-AOC, U/mL	9.15	9.19	9.58	0.19	0.63
d 70					
MDA, nmol/mL	3.38 ^a^	2.48 ^b^	2.30 ^b^	0.15	<0.01
SOD, U/mL	76.34	78.75	81.61	1.53	0.40
GSH-Px, U/mL	173.41 ^b^	177.67 ^b^	210.75 ^a^	5.66	<0.01
CAT, U/mL	53.28 ^b^	57.02 ^ab^	66.39 ^a^	2.14	0.02
T-AOC, U/mL	9.27	9.66	10.36	0.22	0.14

^1^ CON, piglets in the control group fed a basal diet; ZnO, piglets in ZnO group fed a basal diet supplemented with 1500 mg/kg of ZnO; and TBZC, piglets in TBZC group fed a basal diet supplemented with 680 mg/kg of TBZC. ^2^ SEM, standard error of the mean (*n* = 5). ^a, b^ Mean values within a row with different letters differ at *p* < 0.05. And 0.05 ≤ *p* < 0.10 was considered as a tendency.

**Table 6 animals-15-03071-t006:** Effects of TBZC on serum immune indexes of weaned piglets.

Items	CON ^1^	ZnO	TBZC	SEM ^2^	*p*-Value
d 14					
IgA, g/L	0.87	1.06	0.98	0.06	0.52
IgM, g/L	1.74	2.07	1.85	0.11	0.50
IgG, g/L	15.03	17.17	15.62	0.96	0.67
d 28					
IgA, g/L	1.19	1.20	1.52	0.09	0.22
IgM, g/L	2.15	2.22	2.38	0.11	0.70
IgG, g/L	17.69	18.45	20.11	0.99	0.63
d 42					
IgA, g/L	1.56	1.31	1.21	0.08	0.16
IgM, g/L	2.62	2.45	2.24	0.10	0.30
IgG, g/L	20.85	20.87	21.03	0.76	0.99
d 70					
IgA, g/L	1.58	1.68	1.38	0.08	0.28
IgM, g/L	2.83	2.93	2.51	0.10	0.20
IgG, g/L	21.89	21.78	20.55	0.66	0.69

^1^ CON, piglets in the control group fed a basal diet; ZnO, piglets in ZnO group fed a basal diet supplemented with 1500 mg/kg of ZnO; and TBZC, piglets in TBZC group fed a basal diet supplemented with 680 mg/kg of TBZC. ^2^ SEM, standard error of the mean (*n* = 5).

## Data Availability

The data presented in this study are available from the corresponding author on request.

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
