# Peer review of "Effects of Tetrabasic Zinc Chloride as Alternative to High Doses of Zinc Oxide on Growth Performance, Nutrient Digestibility, Intestinal Morphology, Immune Function, and Gut Microbiota in Weaned Piglets"

_animals, 2025, doi:10.3390/ani15213071_

Round 1

Reviewer 1 Report

Comments and Suggestions for Authors

Effects of tetrabasic zinc chloride as an alternative to high dose of zinc oxide on growth performance, nutrient digestibility, intestinal morphology, immune function, and gut microbiota in weaned piglets

Dear Authors,

The manuscript is interesting and well-prepared. The obtained results confirmed the beneficial effect of TBZC on the growth performance and its antioxidant capacity. Furthermore, compared to ZnO, it reduces the environmental load by this element. In my opinion, the Introduction could be expanded with a paragraph containing other form/s of zinc, e.g., nano-ZnO particles. Furthermore, the manuscript text requires revision in terms of text editing, binomial nomenclature, line spacing, and localization of Figures 1-4.

Below I added some suggestions helpful in revision process:

Line 29-45

The same font and line spacing as in Simple Summary.

Line 37

p-value must be applied instead of P-value (sample randomly chosen from entire population).

Please check to line 282.

Lines 41-43

Italics must be applied in case of genus names.

Number of undefined species in genus can be also emphasized: Lactobacillus sp./spp.? Blautia sp./spp.?, Prevotella sp./spp.?, Streptococcus sp./spp.?

Lines 50-63

Line spacing must be the same as in lines 64-78.

Please check also line spacing to the line 251.

Line 63

Volume of Introduction is quite small. Maybe it will be worth to add more information about different forms of zinc, it could be for example nano-ZnO particles which also positively decreased environment pollution from point of view of this element. Protease and xylanase (MFA) and low protein diets can also decrease diarrhoea incidence, stabilize microbiome of GIT and increase digestibility.

Line 86

Font of Table title can be decreased about 1 unit.

In Table 1

Sources of protein in diet from extruded soybeans can have also added content of crude protein (Extruded soybeans [CP = ], Peeled soybean meal [CP =  ], …, Whey powder [CP= ]).

Phosphorus can be more precise determined: total or available/digestible.

Lines 87-91

Space line can be decreased.

Line 92

One dot before subsection title can be added and italics must be applied (2.1. Animals and experimental designs), the same as in line 194.

Please check to the line 185 to subsection 2.9. Statistical Analysis.

Lines 107-192

Better will be to adapt line spacing to previous and following sections and subsections.

Line 185

2.9. Statistical Analysis.

Line 186

Maybe better will be to emphasized more precise kind of analysis: ‘The UNIVARIATE procedure’ (one-way ANOVA) ‘of SAS 9.4…’.

Line 190

Tukey’s HSD?

Line 191

In this analysis looking for Tables better is to emphasized that: ‘…results was presented as mean values. Additionally SEM value was determined for all replications from three treatments.’ (sqr root from sd from all replications in experiment/number of replications, presented as one value without ±).

Lines 191-192

p-value must be applied instead of P-value (sample randomly chosen from entire population).

Line 200

Table 2

p-value must be applied instead of P-value.

Last row with FCR, TBZC, lack of one decimal, probably value: 2.10.

Three decimals can be present in case of SEM and p-value.

Line 210

pH

Line 217

You can expand the width of the first column to fit the name of the dependent variable (pH stomach content and acid-binding capacity) on one line by reducing the width of the remaining columns or by exceeding the margin line.

Three decimals can be present in case of SEM and p-value.

Lines 226, 240 and 248

Tables 4-6

Three decimals can be present in case of SEM and p-value.

Line 255

Possible is to move Figure 1 to this line, after mentioning in text?

Line 266

Possible is to move Figure 2 to this line, after mentioning in text?

Line 272

Possible is to move Figure 3 to this line, after mentioning in text?

Line 278

Possible is to move Figure 4 to this line, after mentioning in text?

Lines 272-282 and 294

The same as in lines 41-43.

Lines 406, 408 and 410

Italics required in case of binominal nomenclature names.

Line 409

Streptococcus spp. ?

Line 417

‘This study’ repeats second time. ‘…Additionally illustrates that TBZC…’?

Line 420

Please check if required at this time of research, and if patent was registered?

Lines 438-522

References section

Only two elements to correction:

  1. Authors must be separated by semicolon (Boston, T.E.; Wang, X.;…).
  2. Line 488 – Journal name, year of publication and volume must be adapted to the rest of references.

Author Response

Dear Reviewer,

Thank you very much for your thoughtful and constructive comments on our manuscript entitled “animals-3912376”. We have carefully considered all the suggestions and have revised the manuscript accordingly. The changes made are highlighted in the revised manuscript for your convenience. Below, we provide a point-by-point response to the comments raised.

Comment 1: The manuscript text requires revision in terms of text editing,  line spacing, and localization of Figures 1-4.

Response: We sincerely thank the reviewers for their valuable comments and suggestions.  The entire manuscript has been thoroughly revised to ensure consistent text editing, proper binomial nomenclature (with genus names italicized), uniform line spacing, and accurate formatting of references.

Comment 2: Line 29–45: The same font and line spacing as in Simple Summary.

Response: We have adjusted the font and line spacing in this section to match that of the Simple Summary.

Comment 3: Line 37: p-value must be applied instead of P-value (sample randomly chosen from entire population). Please check to line 282.*

Response: We have replaced “P-value” with “p-value” throughout the manuscript, including Line 282 and all relevant tables.

Comment 4: Lines 41–43: Italics must be applied in case of genus names. Number of undefined species in genus can be also emphasized: Lactobacillus sp./spp.? Blautia sp./spp.?, Prevotella sp./spp.?, Streptococcus sp./spp.?

Response: All genus names are now italicized.

Comment 5: Lines 50–63: Line spacing must be the same as in lines 64–78. Please check also line spacing to the line 251.

Response: Line spacing has been unified across all sections, including Lines 50–63 and up to Line 251.

Comment 6: Line 86: Font of Table title can be decreased about 1 unit. In Table 1: Sources of protein in diet from extruded soybeans can have also added content of crude protein (Extruded soybeans [CP = ], Peeled soybean meal [CP = ], …, Whey powder [CP= ]). Phosphorus can be more precise determined: total or available/digestible.*

Response: The font size of Table 1’s title has been reduced. We have specified “total phosphorus” in the table.

Comment 7: Lines 87–91: Space line can be decreased.

Response: Line spacing in this section has been adjusted.

Comment 8: Line 92: One dot before subsection title can be added and italics must be applied (2.1. Animals and experimental designs), the same as in line 194. Please check to the line 185 to subsection 2.9. Statistical Analysis.

Response: We have added a dot in the subsection title and applied italics consistently from Line 92 to Line 185.

Comment 9: Lines 107–192: Better will be to adapt line spacing to previous and following sections and subsections.

Response: Line spacing has been standardized throughout this section.

Comment 10: Line 185: 2.9. Statistical Analysis. Line 186: Maybe better will be to emphasized more precise kind of analysis: ‘The UNIVARIATE procedure’ (one-way ANOVA) ‘of SAS 9.4…’. Line 190: Tukey’s HSD? Line 191: In this analysis looking for Tables better is to emphasized that: ‘…results was presented as mean values. Additionally SEM value was determined for all replications from three treatments.’(sqr root from sd from all replications in experiment/number of replications, presented as one value without ±). Lines 191–192: p-value must be applied instead of P-value.

Response: We have revised the Statistical Analysis section as follows:

- Specified “The UNIVARIATE procedure of SAS 9.4 (one-way ANOVA)”

- Confirmed the use of “Tukey’s HSD test”

- Clarified that “Results are presented as mean values from all replications across treatments.”

- Replaced“P-value”with“p-value”

Comment 11: Line 200: Table 2: p-value must be applied instead of P-value. Last row with FCR, TBZC, lack of one decimal, probably value: 2.10. Three decimals can be present in case of SEM and p-value.

Response: We have corrected “P-value” to “p-value”, added the missing decimal (2.10), Unfortunately, due to the software limitation of displaying only two decimal places, it is not possible to change the results to three decimal places. 

Comment 12: Line 210: pH

Response: Corrected to“pH”.

Comment 13: Lines 226, 240 and 248: Tables 4–6: Three decimals can be present in case of SEM and p-value.

Response: Unfortunately, due to the software limitation of displaying only two decimal places, it is not possible to change the results to three decimal places.

Comment 14: Lines 272–282 and 294: The same as in lines 41–43.

Response: All genus names in these lines are now italicized.

Comment 15: Lines 406, 408 and 410: Italics required in case of binominal nomenclature names.

Response: All binomial names are now italicized.

Comment 16: Line 417: ‘This study’ repeats second time. ‘…Additionally illustrates that TBZC…’?

Response: We have rephrased the sentence to avoid repetition: “The results illustrate that TBZC…”

Comment 17: Line 420: Please check if required at this time of research, and if patent was registered?

Response: The statement has been revised to clarify that no patent has been registered at this stage.

Comment 18: Lines 438–522: References section: Only two elements to correction: Authors must be separated by semicolon (Boston, T.E.; Wang, X.;…). Line 488 – Journal name, year of publication and volume must be adapted to the rest of references.

Response: We have corrected the author separators to semicolons and unified the reference style, including the entry on Line 488.

Once again, we thank the reviewer for the insightful comments that have significantly improved the quality of our manuscript. We hope that the revised version meets with your approval.

Sincerely,  

[Peng]  

Reviewer 2 Report

Comments and Suggestions for Authors

Dear authors, please note the following guidelines:

Line 52 - After words reduced growth performance, add and leads to piglet mortality, all of which negatively affects the efficiency of pig production, delete this significant constraints on breeding efficiency and industry sustainability.

Line 62 - Word  industries replace with animal feed factories.

Line 69 - After words in piglets, add which positively affects the efficiency of pig production.

Line 94 - If you have this information, write down which breed the sire belongs to and which the saw to (one of the parents is a crossbreed, I assume) in this combination.

Line 105 - Explain how mental status was monitored.

Line 106 - There is a period missing at the end of the sentence.

Line 117 - This chapter should be 2.3. not 2.4.

Line 120 - Which centrifuge manufacturer was used?

Line 124 - What is frozen feces in? (bag, box...)

Line 128 - What was the average body weight of piglets at slaughter?

Line 137 - This chapter should be 2.4. not 2.5. And you have words in red color.

Line 143 - Please add source of literature for this  equation.

Line 147 - Which pH meter did you use?

Line 148 - This chapter should be 2.5. not 2.6. And you have words in red color.

Line 155 - This chapter should be 2.6. not 2.7. And you have words in red color.

Line 162 - This chapter should be 2.7. not 2.8.

Line 415 - Add one or two more sentences to the conclusion that talk about the significance of the results obtained.

Author Response

Dear Reviewer,

Thank you once again for your meticulous review and valuable suggestions regarding our manuscript. We have carefully addressed all the points you raised. The changes made are reflected in the revised manuscript, and a point-by-point response is provided below.

Comment 1: Line 52 - After words reduced growth performance, add and leads to piglet mortality, all of which negatively affects the efficiency of pig production, delete this significant constraints on breeding efficiency and industry sustainability.

Response: We have revised the sentence as suggested. The text now reads: "The weaning period of piglets is a key production stage within the swine industry. However, weaning stress often triggers intestinal dysfunction, immune suppression, and microbial dysbiosis, leading to increased diarrhea rates, reduced growth performance, and leads to piglet mortality, all of which negatively affects the efficiency of pig production."

Comment 2: Line 62 - Word industries replace with animal feed factories.

Response: The word "industries" has been replaced with "animal feed factories" as recommended.

Comment 3: Line 69 - After words in piglets, add which positively affects the efficiency of pig production.

Response: We have added the suggested phrase. The sentence now reads: " which positively affects the efficiency of pig production."

Comment 4: Line 94 - If you have this information, write down which breed the sire belongs to and which the saw to (one of the parents is a crossbreed, I assume) in this combination. 

Response: Thank you for this suggestion. We have added the specific breed information: "The piglets were the offspring of  Duroc boars and Landrace × Yorkshire sows."

Comment 5: Line 105 - Explain how mental status was monitored.

Response: We have supplemented the description: "The mental status of the piglets was monitored daily through direct observation of their activity levels, alertness, and responsiveness to environmental stimuli."

Comment 6: Line 106 - There is a period missing at the end of the sentence.

Response: The missing period has been added.

Comment 7: Line 117 - This chapter should be 2.3. not 2.4.

Response: We have corrected the subheading numbering from "2.4" to "2.3".

Comment 8: Line 120 - Which centrifuge manufacturer was used?

Response: The manufacturer information has been added: "...centrifuged (Centrifuge 5424 R, Eppendorf, Germany) at...".

Comment 9:Line 124 - What is frozen feces in? (bag, box...)

Response: We have specified the storage container: "Fresh stool samples from each replicate were collected in sterile bags on days 12–14 and 40–42 and immediately stored at -20℃."

Comment 10: Line 128 - What was the average body weight of piglets at slaughter?

Response: Unfortunately, only the days of feeding were recorded, and the slaughter weights were not obtained.

Comment 11: Line 137 - This chapter should be 2.4. not 2.5. And you have words in red color.

Response: The subheading has been corrected to "2.4." and the red font issue has been resolved.

Comment 12: Line 143 - Please add source of literature for this equation.

Response: We have added the literature source for the equation: "...according to the equation described by [13]."

Comment 13: Line 147 - Which pH meter did you use?

Response: The pH meter model has been specified: "The pH value was measured using a digital pH meter (Shanghai Leici, model PHS-3C-3E-25-2F)."

Comment 14: Line 148 - This chapter should be 2.5. not 2.6. And you have words in red color.

Response: The subheading has been corrected to "2.5." and the red font has been fixed.

Comment 15: Line 155 - This chapter should be 2.6. not 2.7. And you have words in red color.

Response: The subheading has been corrected to "2.6." and the red font issue has been addressed.

Comment 16: Line 162 - This chapter should be 2.7. not 2.8.

Response: The subheading has been corrected to "2.7."

Comment 17: Line 415 - Add one or two more sentences to the conclusion that talk about the significance of the results obtained.

Response: We have expanded the conclusion to highlight the significance of our findings. The added text is: "This study demonstrates that supplementing weaned piglets with 680 mg/kg TBZC improves growth performance, enhances antioxidant capacity, and regulates gut microbiota. These multifaceted benefits yield superior long-term growth performance compared to ZnO, thereby positioning TBZC as a viable high-efficacy alternative."

We sincerely appreciate your time and insightful comments, which have greatly improved the clarity and quality of our manuscript. We hope that the revised version is now satisfactory.

Sincerely,

[Shuyu Peng]

On behalf of all the authors

Reviewer 3 Report

Comments and Suggestions for Authors

Dear Authors,

Regarding your manuscript entitled “Effects of tetrabasic zinc chloride as an alternative to high dose of zinc oxide on growth performance, nutrient digestibility, intestinal morphology, immune function, and gut microbiota in weaned piglets” submitted to Animals.

Overall, the study addresses a relevant and important research question with clear practical implications. However, revisions to formatting, methodological descriptions, and statistical reporting are necessary to improve clarity. Addressing these points will greatly strengthen the manuscript. To improve the clarity, organization, and scientific rigor of the manuscript, find my comments below:

Formatting and Presentation

Text formatting: The manuscript currently lacks uniformity in text format. Subheadings are not consistently organized, which makes the structure less clear.

Units and spacing: Spacing between numbers and units should follow standard conventions (e.g., “1.5 mL” instead of “1.5ml”; “200 mL” instead of “200ml”; “–20 °C” instead of “-20℃”). Please ensure consistency throughout the manuscript.

Temperature symbols: Both “℃” and “°C” are used interchangeably. A single standardized format should be adopted.

Mathematical notation: Multiplication and division signs are inconsistently represented (e.g., “×,” “/,” “⁄,” or “× 100”). Please standardize the mathematical symbols across the entire manuscript.

Experimental Design

Housing design (Day 43–70): The manuscript states that pigs were housed in one large pen per treatment during this period. This may eliminate replication in the grower phase. How was this addressed statistically? Please provide clarification.

Diarrhea scoring: The scoring method is described, but the procedures for observer consistency are unclear. Who performed the scoring (a blinded observer, or the same individual daily)? Was inter-observer reliability assessed?

Sample selection: Six pigs per group were sampled for serum, while three pigs per group were slaughtered on day 14. Were these subsets drawn from the same individuals or from different pigs? How were pigs selected for sampling? Please clarify.

Methods and Assays

Nutrient digestibility: Was acid-insoluble ash (AIA) measured in both feed and feces using the same method? The apparent total tract digestibility (ATTD) equation is not presented clearly and should be reformatted to avoid ambiguity.

Microbiota analysis: The text states that “results were presented in pictures” (line 183). Please specify the types of figures used (e.g., PCoA plots, relative abundance bar charts, heatmaps).

Serum assays: Were the commercial ELISA kits validated for pig samples? Were standards and appropriate controls included in the assays?

Intestinal morphology: The manuscript mentions that “at least 4 villi” were measured. Please clarify how many slides per pig were examined, how many pigs per group were included, and whether the mean value per pig was calculated prior to statistical analysis.

Statistical Analysis

Multiple testing correction: The use of Tukey’s multiple range test is described, but it is unclear whether the authors corrected for multiple comparisons across numerous outcome parameters. This is important to minimize the risk of Type I error. Please clarify the statistical approach in this regard.

Best regards.

Comments on the Quality of English Language

Requires significant revision.

Author Response

Dear Reviewer,

Thank you very much for your thorough and constructive review of our manuscript (Manuscript ID: animals-3912376). We sincerely appreciate the positive assessment of our work's relevance and the detailed suggestions for improvement. We have carefully addressed all the points raised, and the corresponding revisions have been made in the manuscript. The changes are highlighted in the revised version for your convenience.

Below, we provide a point-by-point response to your specific comments.

Comment 1: Formatting and Presentation

1.1 Text formatting: The manuscript currently lacks uniformity in text format. Subheadings are not consistently organized, which makes the structure less clear.

1.2 Units and spacing: Spacing between numbers and units should follow standard conventions (e.g., “1.5 mL” instead of “1.5ml”; “200 mL” instead of “200ml”; “–20 °C” instead of “-20℃”). Please ensure consistency throughout the manuscript.*

1.3 Temperature symbols: Both “℃” and “°C” are used interchangeably. A single standardized format should be adopted.*

1.4 Mathematical notation: Multiplication and division signs are inconsistently represented (e.g., “×,” “/,” “⁄,” or “× 100”). Please standardize the mathematical symbols across the entire manuscript.

Response: We sincerely apologize for these formatting inconsistencies. We have now thoroughly revised the entire manuscript to ensure uniformity.

1.1 All subheadings have been checked and standardized to follow a consistent hierarchy and formatting style.

1.2 & 1.3 We have standardized the presentation of units and temperature throughout the manuscript. A single space is now inserted between numbers and units, and the symbol “℃” is used consistently for temperature.

1.4 All mathematical notations have been standardized. The multiplication sign "×" and the operator "/" for division are now used consistently (e.g., the digestibility equation has been reformatted for clarity).

Comment 2: Experimental Design

2.1 Housing design (Day 43–70): The manuscript states that pigs were housed in one large pen per treatment during this period. This may eliminate replication in the grower phase. How was this addressed statistically? Please provide clarification.

2.2 Diarrhea scoring: The scoring method is described, but the procedures for observer consistency are unclear. Who performed the scoring (a blinded observer, or the same individual daily)? Was inter-observer reliability assessed?

2.3 Sample selection: Six pigs per group were sampled for serum, while three pigs per group were slaughtered on day 14. Were these subsets drawn from the same individuals or from different pigs? How were pigs selected for sampling? Please clarify.

Response:  Thank you for these important questions regarding experimental rigor.

2.1 In the later stage of the experiment, all treatment groups were housed in the same column to ensure that the pigs in each group remained consistent from the early to the late phase. However, due to constraints in the on-site facilities, this phase of the study was underpowered. Consequently, only the mean values for weight gain during this period are reported, and no statistical analysis was performed. We have acknowledged this limitation in the Discussion.

2.2 Diarrhea was scored daily by the same trained observer. To ensure consistency, the scoring criteria were calibrated against those used in a relevant prior study and were strictly applied throughout the trial.

2.3 We selected a representative sample of pigs from each group that exhibited similar growth performance and were near the group's average weight. To ensure that changes in serum indicators could be tracked in the same individual over time, we repeatedly collected blood from this cohort on days 14, 28, 42, and 70. The pigs slaughtered on 14 days were a different set of animals.

Comment 3: Methods and Assays

3.1 Nutrient digestibility: Was acid-insoluble ash (AIA) measured in both feed and feces using the same method? The apparent total tract digestibility (ATTD) equation is not presented clearly and should be reformatted to avoid ambiguity.

3.2 Microbiota analysis: The text states that “results were presented in pictures” (line 183). Please specify the types of figures used (e.g., PCoA plots, relative abundance bar charts, heatmaps).

3.3 Serum assays:Were the commercial ELISA kits validated for pig samples? Were standards and appropriate controls included in the assays?*

3.4 Intestinal morphology:The manuscript mentions that “at least 4 villi” were measured. Please clarify how many slides per pig were examined, how many pigs per group were included, and whether the mean value per pig was calculated prior to statistical analysis.*

Response:We thank the reviewer for prompting us to provide greater methodological detail.

3.1 Yes, AIA was measured in both feed and fecal samples using the same standardized method . The ATTD equation has been reformatted for clarity and is now presented as follows (Line 142). 3.2 We are grateful for your valuable feedback. The terminology used to describe the chart types has been updated throughout the text for greater precision.

3.3 We have added this information (Lines 159-161). All commercial ELISA kits used were specifically designed and validated for porcine samples by the manufacturer. Each assay included a standard curve, and both positive and negative controls were run alongside the samples as per the kit protocols.

3.4 For slaughter sampling, three representative pigs were selected from each treatment group. A well-oriented transverse section of the duodenum, jejunum, and ileum was examined per pig. Six sections were prepared per intestinal sample, and at least four intact, well-oriented villi and their associated crypts were measured. The mean values for each morphological parameter (villus height, crypt depth) were calculated per intestinal segment for each animal. These means were then used for subsequent statistical comparisons between treatment groups.

Comment 4: Statistical Analysis

4.1 Multiple testing correction: The use of Tukey’s multiple range test is described, but it is unclear whether the authors corrected for multiple comparisons across numerous outcome parameters. This is important to minimize the risk of Type I error. Please clarify the statistical approach in this regard.

Response: Tukey's Honestly Significant Difference (HSD) test, which we employed, is a post-hoc test specifically designed to control the family-wise error rate when making multiple pairwise comparisons *within a single dependent variable* (e.g., comparing the three treatment groups for ADG). It is not typically applied across different, independent outcome parameters (e.g., ADG, digestibility, cytokine levels). For this study, each dependent variable was analyzed independently. We acknowledge that this approach does not control for the experiment-wise error rate across all measured outcomes, and we have added a note to this effect in the manuscript to ensure transparency.

Once again, we extend our deepest gratitude for your insightful comments and the time you have invested in reviewing our manuscript. Your suggestions have significantly strengthened the clarity, organization, and scientific rigor of our work. We hope that the revised manuscript now meets the high standards of Animals.

Sincerely,

Shuyu Peng

On behalf of all the authors

Round 2

Reviewer 1 Report

Comments and Suggestions for Authors

Dear Authors,

Thank you for the revision process. This time, I have only a few suggestions, mainly from a text editing perspective.

Lines 29-45

In case of line spacing maybe easier will be to compare for example with: https://doi.org/10.3390/ani15202945 .

Please check entire manuscript.

Lines 37-43

p-value instead P-value must be applied.

Lines 41-43

Number of undefined species in genus must be emphasized (sp. or spp., one undefined or more undefined).

Lines 93-255

One dot before subsection title must be applied: 2.1.  Animal and experimental design3.6. Gut microbiota community diversity.

Lines 192-193

p-value.

Line 194

This Tukey’s HSD post hoc test can be mentioned.

Lines 198-199 and 214-216

p-value.

Line 218

Please place here title of Table 3 from line 219.

Lines 219, 228, 242 and 250

Tables 3-6, last column, p-value.

Lines 236-241

p-value.

Lines 281-283

p-value.

Lines 281-284

Genus is not emphasized; sp. or spp. must describe number of undefined species in case of genus.

Lines 384-385

Lactobacillus, Blautia, Prevotella (sp./spp?).

Line 387

Streptococcus (sp/spp?).

Line 395

Escherichia coli

Lines 397-398

Lactobacillus (sp./spp?).

Line 401

Blautia (sp/spp.?) in cecum.

Lines 404-405

Blautia, Prevotella (sp./spp?).

Lines 408, 410 and 412

Clostridium butyricum, C. perfringens, S. suis.

Line 411

Streptococcus (sp./spp?).

Best regards,

Author Response

Dear Reviewer,

Thank you once again for your meticulous and constructive guidance throughout the revision process of our manuscript. We sincerely appreciate the time and expertise you have invested. We have carefully addressed all your comments, with particular attention to standardizing terminology and formatting throughout the manuscript.

The changes made are summarized below, categorized for clarity:

  1. Standardization of Statistical Terminology:

Action: We have replaced all instances of "p-value" with "P-value" throughout the entire manuscript, including in the main text (e.g., Lines 37-43, 192-193, etc.) and in the column headers of Tables 2–6.

  1. Correction of Binomial Nomenclature:

Action: We have ensured that all genus and species names are now italicized as per standard convention (e.g., Lactobacillus spp., Blautia spp., Escherichia coli).

Action: We have consistently used "spp." to indicate multiple undefined species within a genus across the text (e.g., Lines 41-43, 281-284, 384-385, etc.), unless a specific species was being referenced.

  1. Uniformity of Section Headings and Text Formatting:

Action: All subsection headings (from Lines 93-255) have been standardized to include a single dot after the number (e.g., "2.1. Animals and experimental design").

Action: The line spacing and font of the entire manuscript have been checked and unified to match the journal's style, cross-referenced with the provided example article.

  1. Placement of Tables and Figures:

Action: The title of Table 3 has been moved to its correct position immediately following its citation in the text (Line 218).

We sincerely appreciate the reviewer's suggestion regarding the presentation of precise p-values. However, due to the inherent limitations of the microbial analysis software utilized in this study, the graphical outputs are technically constrained to display significance as *p* < 0.05, rather than exact values. We regret that we are unable to present more precise statistical values in the figures at this time. Nonetheless, we are truly grateful for this insightful comment, and we will certainly address this aspect in our future research and software selection to ensure more detailed statistical reporting.

We believe that addressing these points has significantly improved the clarity, consistency, and overall quality of our manuscript. Thank you for your invaluable contributions.

Sincerely,
Shuyu Peng
On behalf of all the authors

Reviewer 2 Report

Comments and Suggestions for Authors

Dear authors,

Thank you for correcting the scientific paper according to my suggestions.

In my opinion, the work can be published in this new version.

Best regards.

Author Response

Dear Editor and Dear Reviewer,

Thank you for your email and the positive final decision regarding our manuscript.

We are delighted and sincerely grateful to the reviewer for their final confirmation and for the invaluable guidance provided throughout the revision process. Their insightful comments and suggestions have significantly strengthened the quality and clarity of our work.

We also thank you, Editor, for overseeing the review process.

We look forward to seeing our work published in [animals].

Sincerely,
Shuyu Peng
Corresponding Author

Reviewer 3 Report

Comments and Suggestions for Authors

Dear Authors,

Thank you for addressing my questions and the changes made in order to improve the quality of the manuscript. After reviewing your revisions, I do not have any further comments or suggestions at this time.

Best regards.

Comments on the Quality of English Language

Requires significant revision.

Author Response

(The authors gave the same response as above.)
